# A Clustering-Enhanced Memetic Algorithm for the Quadratic Minimum Spanning Tree Problem

**DOI:** 10.3390/e25010087

**Published:** 2022-12-31

**Authors:** Shufan Zhang, Jianlin Mao, Niya Wang, Dayan Li, Chengan Ju

**Affiliations:** Faculty of Information Engineering and Automation, Kunming University of Science and Technology, Kunming 650500, China

**Keywords:** quadratic minimum spanning tree problem, memetic algorithm, combinatorial optimization, agglomerative clustering

## Abstract

The quadratic minimum spanning tree problem (QMSTP) is a spanning tree optimization problem that considers the interaction cost between pairs of edges arising from a number of practical scenarios. This problem is NP-hard, and therefore there is not a known polynomial time approach to solve it. To find a close-to-optimal solution to the problem in a reasonable time, we present for the first time a clustering-enhanced memetic algorithm (CMA) that combines four components, i.e., (i) population initialization with clustering mechanism, (ii) a tabu-based nearby exploration phase to search nearby local optima in a restricted area, (iii) a three-parent combination operator to generate promising offspring solutions, and (iv) a mutation operator using Lévy distribution to prevent the population from premature. Computational experiments are carried on 36 benchmark instances from 3 standard sets, and the results show that the proposed algorithm is competitive with the state-of-the-art approaches. In particular, it reports improved upper bounds for the 25 most challenging instances with unproven optimal solutions, while matching the best-known results for all but 2 of the remaining instances. Additional analysis highlights the contribution of the clustering mechanism and combination operator to the performance of the algorithm.

## 1. Introduction

The quadratic minimum spanning tree problem (QMSTP) is widely used in distributed network design when the interaction between edges must be considered, rather than only the contribution of each individual edge [1]. For example, in the wireless telecommunication network, it is necessary to take into account the cost of the interference between edges sharing the same radio frequency [2]. A similar situation can arise in other cases, such as energy distribution and chip design [3].  QMSTP is an extension of the classic spanning tree optimization problem, the objective of which is to minimize the total cost of a subgraph that connect all vertices in a finitely connected graph, where the total cost is the sum of weights of the edges (linear cost) and the pair of edges (quadratic cost) in that subgraph (a tree).

A special case of QMSTP is the adjacent-only QMSTP (AQMSTP), where the interference costs are limited to pairs of adjacent edges (e.g., bending losses in T-joints in utility networks and turn penalties in transportation) [2,4]. In addition, QMSTP has some related variants including the quadratic bottleneck spanning tree problem and the minimum spanning tree problem with conflict pair constraints [1]. Majumder et al. [5] present a rough fuzzy chance-constrained model and report solutions under high uncertainty. Some special cases of QMSTP are linearizable, and the characterization of such cases is described in [6].

### 1.1. Literature Review

Since Assad and Xu [7] proposed QMSTP in 1992, it has received appreciable attention in the past three decades. A variety of approaches have been proposed for QMSTP, which can be classified into three categories:Exact algorithms: These algorithms can theoretically find a global optimal solution. For instance, Assad and Xu [7] proposed an accurate branch-and-bound algorithm and gave the optimal solution for cases with a maximum number of vertices of 15. Cordone et al. [8] improve the Lagrangian branch-and-bound algorithm in [7]. Pereira et al. [9] introduced two parallel branch-and-bound algorithms embedded with two lower bounding procedures based on Lagrangian relaxation and solved instances, including some with 50 vertices. Rostami and Malucelli [10] developed new lower bounds for instances with up to 50 vertices based on a reformulation scheme and some new mixed 0–1 linear formulations. Exact algorithms are also presented for related QMSTP variants. Guimarães et al. [11] investigated semidefinite programming lower bounds and introduced a new branch-and-bound algorithm, which stands out as the best exact algorithm. Sotirov et al. [12] reported a hierarchy of lower bounds and derived a sequence of improved relaxations with increasing complexity. However, since QMSTP is an NP-hard problem [7], the search space grows exponentially with the number of vertices. With the expansion of the graph, an accurate branch and bound algorithm [7] and its improved algorithms [8,9,10,11,12] deteriorate rapidly.Population-based algorithms: These algorithms perform searches with multiple initial points in a parallel style and associated search operators [13]. Zhou et al. [14] proposed a genetic algorithm using Prüfer number coding, which increased the number of vertices that can be solved to 50 for the first time. However, Prüfer number coding can only be applied to complete graphs and lacks local search capability as well as heritability. For this reason, Palubeckis et al. [3] used edge-set coding to implement a hybrid genetic algorithm with neighborhood search. Sundar et al. [15] proposed an artificial bee colony algorithm (ABC), which assigned employed bees to exploit the neighborhood of a food resource, onlooker bees to explore nearby food resources and scout bees to introduce new food resources, and finally applied local search to further optimize the solution.Local search algorithms: These algorithms starts with a single initial solution and use iterative local search to obtain better solutions. Öncan et al. [16] proposed a local search algorithm that alternates between local search and random moves. Palubeckis et al. [3] proposed an iterative tabu search (ITS) algorithm and achieved the best results in comparison with multistart simulated annealing algorithm and hybrid genetic algorithm on instances introduced by Cordone and Passeri [17]. Cordone et al. [8] applied the improved data structure and neighborhood search to the tabu search algorithm and started a new round of search with a randomly generated new solution after several iterations. The algorithm HSII proposed by Lozano et al. [18] combines iterative tabu search and oscillation strategy, and its neighborhood search is characterized by preferentially replacing edges that contribute the most to the current solution cost. Fu et al. [19] proposed a three-phase search approach (TPS), which includes a descent neighborhood search phase to reach a local optimum, a local optima exploring phase to discover nearby local optima within a given regional area and a perturbation-based diversification phase to jump out of the current regional search area. The algorithm achieved the best results in the literature.

Population-based approaches [3,13,14,15] combine both the population-based framework and neighborhood search, but the process of generating a new solution for neighborhood search in [3] only relies on the crossover operator, thus making the exploration capability weak, and the neighborhood search in [14] is only used for final refinement after the entire exploration process is over, thus making the exploitation insufficient. On the other side, local search algorithms [3,8,17,18,19] show a lack of utilization of global information, resulting in the unnecessary waste of exploitation cost.

The memetic algorithm [20] is an effective metaheuristic framework that combines the global exploration of population-based algorithms and the neighborhood exploitation of local search algorithms. MA has proved its effectiveness in solving discrete optimization problems [21,22]. However, there is currently no application of MA in QMSTP, so in this work we fill the gap.

### 1.2. Contribution

In this paper, we present an effective clustering-enhanced memetic algorithm (CMA) that attains high-quality solutions to QMSTP. The main contributions are as follows:
First, we introduce a clustering mechanism to the edges of the graph, where edges can be grouped with short “distance” (total cost) in the same cluster. This mechanism can effectively guide the population initialization. Second, we adopt a tabu-based nearby exploration phase to explore the restricted surrounding areas to locate other local optima, which can guarantee the search intensification around chosen high-quality local optima. Third, we design a new combination operator and a new mutation operator. By inheriting valuable edges, the combination operator can generate promising and diversified offspring solutions that are used as starting points for local refinement. The mutation operator, which is designed based on Lévy distribution, can explore unvisited search regions, and it can prevent the population from premature.  Finally, we integrate the above three parts as an overall solution, where all parts coordinate and promote each other to find the optima. Additionally, CMA reports improved upper bounds for the 25 most challenging benchmark instances with unknown optimal solutions, which are valuable references for future research on this problem.

## 2. Problem Description

Consider a connected undirected graph G=(V,E) with |V|=n vertices and |E|=m edges, linear costs $c_{e}$ are defined on each edge e∈E and quadratic costs $q_{ef}$ are defined on each pair of edges e,f∈E. The quadratic minimum spanning tree problem is to determine a spanning tree S=(V,T) that minimizes the sum of the linear cost of all edges in *T* plus the quadratic cost of all pairs of edges in *T*. Without loss of generality, we assume $q_{ef}=q_{fe}$ for all e,f∈E, and $q_{ee}=0,e\in E$. The quadratic minimum spanning tree problem is formulated as follows:objective function:(1)$$\min f\left(T\right)=\sum_{e\in E}c_{e}X_{e}+\sum_{e\in E}\sum_{f\in E}q_{ef}X_{e}X_{f}$$subject to: 
(2)$$\begin{array}{c} \sum_{e\in E}X_{e}=n-1 \end{array}$$
(3)$$\begin{array}{c} \sum_{e\in E(P)}X_{e}\leq \left|P\right|-1,\forall P\subset V\;\mathrm{with}\;|P|\geq 3 \end{array}$$
(4)$$\begin{array}{c} X_{e}\in \left\{0,1\right\},e\in E \end{array}$$
where if the edge *e* belongs to the solution *T*, then $X_{e}=1$, otherwise $X_{e}=0$. P(|P|≥3) is any vertex set of *V*, and E(P) represents the set of edges whose both endpoints belong to *P*. Equation (Equation 2) requires that the final solution contains n−1 edges, and Equation (Equation 3) ensures that no loops are formed in the solution. These constraints jointly guarantee that the obtained solution must be a spanning tree.

## 3. Memetic Algorithm for the Quadratic Minimum Spanning Tree Problem

### 3.1. Main Scheme

The general architecture of the proposed CMA for QMSTP is summarized in Algorithm 1. The basic idea behind the proposed algorithm is to alternate between the diversification of the population to explore the global promising search space and the dedicated exploitation of every new solution to find refined local optima. Precisely, the algorithm starts with an edge partition based on agglomerative clustering which divides the edge set into several edge pools for initial population generation. Then, it updates the population by iteratively applying the combination/mutation operator to generate new solutions and a dedicated exploitation procedure to further improve every attempt, followed by a population updating procedure to maintain the advancement and diversification of the population, until the stop condition is met. A detailed description of the algorithmic components is provided in the following sections.
**Algorithm 1:** General architecture of CMA for the QMSTP
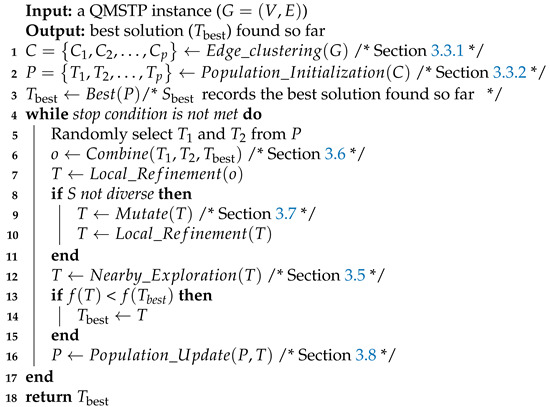


### 3.2. Solution Representation

Each solution *S* is represented by a permutation $F=\left\{f_{1},f_{2},\ldots ,f_{i}\right\}\left(i=n\right)$, where each $f_{i}$ denotes the parent vertex of vertex *i*, except the vertex 1, which is fixed as the root vertex ($f_{1}=Null$). Figure 1 gives an example of a tree and its representation. Given a representation, the complete edge set of the tree can be obtained by traversing the permutation *F*, in which $\left[i,f_{i}\right]\left(i\in V\right)$ represent one single edge.

### 3.3. Population Initialization with Edge Clustering Mechanism

The proposed algorithm starts its search with a population of promising diversified solutions, which is obtained by two steps. The first step is the partition of the edge set with the edge clustering mechanism (Section 3.3.1), and the second is the initial solution’s generation (Section 3.3.2) based on the partitions obtained.

#### 3.3.1. Edge Clustering Mechanism

A quadratic minimum spanning tree minimizes the total cost of a cluster of edges which the tree consists of. This feature is similar to agglomerative clustering, which considers the cost as similarity. Thus, finding a cluster of edges with higher similarity is helpful to bring up a tree with lower total cost.

Agglomerative clustering is an unsupervised classification mechanism which partitions a set of objects into subsets (called clusters) such that each subset contains similar objects, and objects in different subsets are dissimilar on the basis of a similarity metric [23,24]. At the beginning, every input object forms its own cluster. In each subsequent step, the two “closest” clusters will be merged until a given criterion is satisfied [23].

We adopt the pairwise distance to build the similarity metric, so that two clusters with a small distance should have a large similarity. Let *C* denote cluster vector, *D* denote distance matrix. The distance $D_{ij}$ between cluster $C_{i}$ and cluster $C_{j}$ is defined as the potential cost of a tree spanning from the new cluster $C_{i}\cup C_{j}$. According to Equation (Equation 1), the $D_{ij}$ is given as:(5)$$D_{ij}=\frac{\sum_{e\in C_{i}\cup C_{j}}w_{e}}{m_{C_{i}}+m_{C_{j}}}+\frac{\left(\sum_{e\in C_{i}\cup C_{j}}\sum_{f\in C_{i}\cup C_{j}}q_{ef}\right)\cdot \left(m-1\right)}{\left(m_{C_{i}}+m_{C_{j}}\right)\cdot \left(m_{C_{i}}+m_{C_{j}}-1\right)}$$
where $w_{e}$ is used to replace $c_{e}$ in Equation (Equation 1) (while both representing the linear cost of edge *e*) in order to avoid conflict expression with cluster $C_{e}$, and $m_{C_{i}},m_{C_{j}}$ denote the number of edges in $C_{i},C_{j}$, while *m* represents the number of edges of the wanted tree. The derivation process of this equation is as follows: considering a tree with *m* edges, then the total cost will consist of *m* items of linear costs and m(m−1) items of quadratic costs. This means that we can use the average linear cost $w_{\mathrm{ave}}$ and the average quadratic cost $q_{\mathrm{ave}}$ to figure out the total cost as $m\times w_{\mathrm{ave}}+m\left(m-1\right)\times q_{\mathrm{ave}}$, and we can further omit an *m* from it in comparison in a same instance, which makes it $w_{\mathrm{ave}}+\left(m-1\right)\times q_{\mathrm{ave}}$. We can easily obtain the $w_{\mathrm{ave}}$ of two clusters $C_{i},C_{j}$ by dividing the sum of all linear costs in $C_{i}\cup C_{j}$ by $m_{C_{i}}+m_{C_{j}}$, the total number of items of them. Similarly, $q_{\mathrm{ave}}$ can be obtained by dividing the sum of all quadratic costs in $C_{i}\cup C_{j}$ by $\left(m_{C_{i}}+m_{C_{j}}\right)\cdot \left(m_{C_{i}}+m_{C_{j}}-1\right)$, the total number of items of them. In this way, Equation (Equation 5) is obtained.

In order to speed up the process of clustering, we optimize the data structure and the update procedure. Let *Q* denote the quadratic cost matrix between clusters, then *Q* is an upper triangular matrix, which the quadratic cost between $C_{i}$ and $C_{j}$ only stored in $Q_{C_{i}C_{j}}$ supposes that i<j. Let $w_{c_{i}},w_{c_{j}}$ denote the linear cost of edges in $C_{i},C_{j}$ and $q_{C_{i}},q_{C_{j}}$ denote the quadratic cost among edges in $C_{i},C_{j}$, respectively. Then, Equation (Equation 5) can be stated as:(6)$$D_{ij}=\left(w_{C_{i}}+w_{C_{j}}\right)\cdot \frac{1}{m_{C_{i}}+m_{C_{j}}}+\left(q_{C_{i}}+q_{C_{j}}+Q_{C_{i}C_{j}}\right)\cdot \frac{\left(m-1\right)}{\left(m_{C_{i}}+m_{C_{j}}\right)\cdot \left(m_{C_{i}}+m_{C_{j}}-1\right)}$$
where $w_{c_{i}},q_{c_{i}}$ can be stored as property in the $C_{i}$. If $C_{i}$ and $C_{j}$ merge into new $C_{i}$, the property of $C_{i}$ can be updated simply as below: (7)$$w_{C_{i}}=w_{C_{i}}+w_{C_{j}}$$
(8)$$q_{C_{i}}=q_{C_{i}}+q_{C_{j}}+q_{C_{i}C_{j}}$$

Additionally, the *Q* only needs to update partially, as below: (9)$$Q_{C_{k}C_{i}}=Q_{C_{k}C_{i}}+Q_{C_{k}C_{j}},k<i,j$$
(10)$$Q_{C_{i}C_{k}}=Q_{C_{i}C_{k}}+Q_{C_{k}C_{j}},i<k<i$$
(11)$$Q_{C_{i}C_{k}}=Q_{C_{i}C_{k}}+Q_{C_{j}C_{k}},k>i,j$$

Based on the above similarity measurement, the edge clustering process is shown in Algorithm 2. Given a clustering ratio *r*, once the size of generated cluster is greater than or equal to n·r (just suitable for constructing a tree with *n* vertices), it will be returned as a resulting cluster. The process goes on until there are not enough edges for making up a new cluster. Owing to the problem-specific similarity measurement, the clustering process can be finished in $O(m^{2})$ time, where *m* is the number of edges in the graph.
**Algorithm 2:** Edge clustering
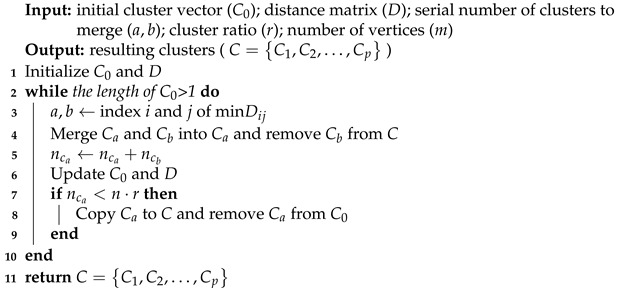


Figure 2 shows an example of the agglomerative clustering process. Given a graph consisting of 9 edges $\left\{e_{i},i=1,\ldots ,9\right\}$, we regard every edge as an initial cluster and create the initial distance matrix of them. Then, we find the “closest” pair of clusters (presuming $\left\{e_{1}\right\}$ and $\left\{e_{5}\right\}$), merge them into a new cluster and update the distance matrix using Equations (Equation 5)–(Equation 11). Then, we iteratively find the next pair of clusters to merge, for example, they are $\left\{e_{3}\right\}$ and $\left\{e_{4}\right\}$, $\left\{e_{6}\right\}$ and $\left\{e_{8}\right\}$, $\left\{e_{1},e_{5}\right\}$ and $\left\{e_{7}\right\}$, $\left\{e_{3},e_{4}\right\}$ and $\left\{e_{2}\right\}$, $\left\{e_{1},e_{5},e_{7}\right\}$ and $\left\{e_{9}\right\}$, in sequence. When it comes to the cluster $\left\{e_{1},e_{5},e_{7},e_{9}\right\}$, the cluster is big enough (assuming the criterion n·r=4) to be returned as a resulting cluster, and the remaining clusters will continue the process until there are not enough edges for making up another new cluster (in this example, another cluster $\left\{e_{2},e_{3},e_{4},e_{6},e_{8}\right\}$ can be returned as well).

#### 3.3.2. Initial Solution Generation

We use a simple procedure inherited from [19] to build a feasible spanning tree *T* for each cluster *C*. Initialize empty solution *T* with root vertex, and iteratively select an edge *e* from *C* to construct *T*. The chosen edge should satisfy such a requirement that one vertex of it belongs to *T*, while another does not. If such an edge cannot be found in *C*, then it will turn to *E* for complement. The procedure will go on until a feasible solution T=(V(T),E(T)) is completely constructed, where $E\left(T\right)=\left\{e_{1},e_{2},\ldots ,e_{n-1}\right\}$ and $V\left(T\right)=\left\{v_{1},v_{2},\ldots ,v_{n}\right\}$ denote the edge set and vertex set of *T*, respectively. For all clusters $C_{i},i=1,2,\ldots ,p$, we construct corresponding initial solutions $T_{i},i=1,2,\ldots ,p$, which make up the initial population. This process is performed in O(n) time, where *n* is the number of vertices of graph G=(V,E). The initial population generation process is shown in Algorithm 3.
**Algorithm 3:** Initial population generation from edge clusters
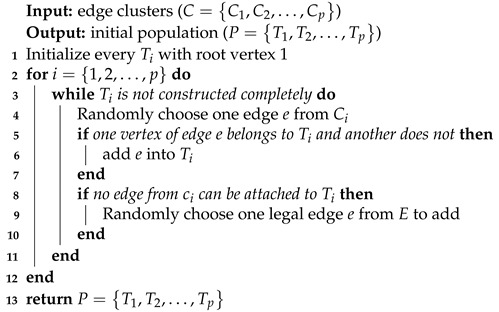


### 3.4. Local Refinement

A QMST solution is possible to be improved by changing only one of its edges. To realize such an improvement, several local refinement algorithms have been proposed [3,8,17,18,19]. In local refinement, the move operator determines the set of candidate solutions where the search looks for improvement. According to comparative tests, our CMA adopts an efficient move operator to realize the descent local refinement procedure along with an iterative mechanism. Local refinement keeps transitioning the current solution to a better state, and finally locks on to the local optimum, as described in Algorithm 4.
**Algorithm 4:** Local refinement
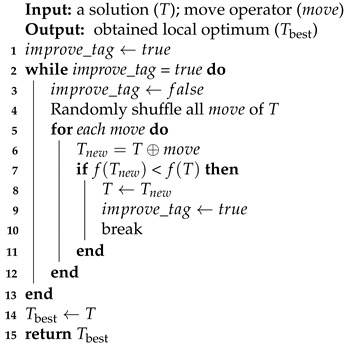


#### 3.4.1. Move Operator

Let $E\left(T\right)=\left\{e_{1},e_{2},\ldots ,e_{n-1}\right\}$ and $V\left(T\right)=\left\{v_{1},v_{2},\ldots ,v_{n}\right\}$ denote the edge set and vertex set of a tree *T*, respectively. We randomly remove an edge e∈E(T) from the current solution *T*, thus partitioning the tree into two subtrees $T^{\prime }$ and $T^{\prime \prime }$. Then, select an edge $f\in V\left(T^{\prime }\right)\times V\left(T^{\prime \prime }\right)$ from the set of edges that connect $T^{\prime }$ and $T^{\prime \prime }$ to recombine these two part into a new feasible solution $T_{\mathrm{new}}$, with $E\left(T_{\mathrm{new}}\right)=\left(E\left(T\right)\setminus \left\{e\right\}\right)\cup \left\{f\right\}$ subject to Equations (Equation 2)–(Equation 4) (see example in Figure 3). We use $T_{\mathrm{new}}=T⊕\mathrm{move}\left(e,f\right)$ to indicate the resulting solution $T_{\mathrm{new}\;}$ acquired by applying move(e,f) to *T*. The amount of possible move(e,f) is bounded by $O(n^{2})$. This move operator is also used in tabu-based nearby exploration (Section 3.6) and mutation operator (Section 3.7). Figure 3 shows the process of the move operator: (a) represents the original solution; in (b), an edge connecting vertex 2 to vertex 4 is removed; (c) represents all the possible edges that can be added; and (d) is an edge that connect vertex 2 to vertex 3 that is chosen to be added, thus making it a legal solution again.

#### 3.4.2. Evaluation Technique

In order to speed up the evaluation process, we adopt a dedicated technique inherited from [8,19]. We use a vector *D* to store the actual or potential contribution of each edge e∈T to the overall cost of the current solution *T*, shown as Equation (Equation 12).
(12)$$D_{g}=c_{g}+\sum_{f\in E(T)}\left(q_{gh}+q_{hg}\right),g\in E$$

When performing move(e,f), the variation in cost is given by Equation (Equation 13), which can be calculated in O(1) time. After executing the picked move(e,f), the whole vector *D* is updated by Equation (Equation 14) in O(m) time, where *m* is the number of edges in graph G=(V,E).
(13)$$\delta_{ef}=D_{e}-D_{f}-q_{ef}-q_{fe}$$
(14)$$D_{g}=D_{g}-q_{ge}-q_{eg}+q_{gf}+q_{fg}$$

### 3.5. Tabu-Based Nearby Exploration

In the search space surrounding an original local optimum, there may be other better local optima. In order to find such nearby local optimum of solutions in the population, we present a tabu-based nearby exploration which is based on a tabu search (Section 3.5.1) to transition the current solution to a nearby region and local refinement (Section 3.4) for nearby local optimum acquiring. A distinguished feature of the exploration is that every time a new local optimum is obtained, it will be compared with original local optimum, and the better one will be the new starting solution of the next iteration. The exploration alternately applies tabu search and local refinement to find nearby local optima until the starting solution has not been updated for $c_{\max}$ consecutive times, as described in Algorithm 5.
**Algorithm 5:** Tabu-based nearby exploration
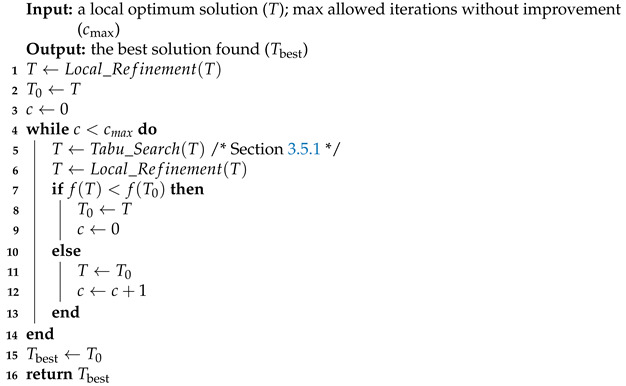


#### 3.5.1. Tabu Search

Tabu search has been successfully applied to solve QMSTP in the literature [3,8,17,18,19]. It allows both improving and nonimproving moves to jump out of the original local optimum and use the tabu mechanism to avoid the search from entering the detour (returning to the solution that has been searched). Our tabu search applies the move operator (Section 3.4.1) but differs from local refinement in the selection of replacement edges. For one move, tabu search arbitrarily selects an edge *e* which is not in the tabu list (Section 3.5.2) to delete without any other attempt and selects one edge from the set of feasible edges with the smallest degradation to add. This move will be iteratively performed $l_{e}$ times, which is regarded as the exploration length. The time complexity of tabu search is bounded by $O(n^{2})$.

#### 3.5.2. Tabu List Management

To prevent from revisiting the same solution encountered just several moves before, each edge that is added into the current solution *T* is forbidden to be removed in the following next $t_{out}$ moves (tabu tenures). The reason why there is no tabu list for edges to add is that experimental tests show that the probability of selecting the edge just removed to add is quite small. In addition, the tabu restriction of a move is ignored if the move leads to an improvement of the starting local optimum solution (aspiration criterion).

### 3.6. Three-Parent Combination Operator

The combination operator is a driving force of memetic algorithms [20] which is able to generate new solutions, not only inheriting the good elements of their parents but also differing from them. Here, we propose a three-parent combination operator to QMSTP for the first time to recombine three existing solutions in the population, one of which is the best individual, while the other two are selected at random. The proposed operator works to reconstruct a new tree as an offspring solution from scratch based on the edge sets of parents. Precisely, the combination operator starts with an empty solution only consisting of the root vertex and then iteratively selects a parent solution from the above three to provide an edge for tree construction. The probability to choose the best individual is $p_{b}$ and $(1-p_{b})/2$ to choose one of the other two. If there is no feasible edge that can be selected in the chosen parent, the operator will turn to other parents for supplement. As the example shows in Figure 4, subgraphs (g) to (i) represent three parents, where the arrow style represents corresponding edges of them, and the generation of an offspring solution is depicted in (a) to (f). Subgraph (a) means that the first parent chosen to provide an edge for tree construction is parent A, and the chosen edge from parent A is the edge connecting vertex 5 to root vertex 1 (in this case, this edge is the only choice). Subgraph (b) means the next chosen edge to be attached to the offspring in progress is from parent B. In this way, we consecutively choose a parent from parent (A, B, C) to provide an edge to be attached to the current incomplete tree until the tree is completed, as shown in subgraph (f).

The proposed operator aims to guarantee guidance for offspring in the optimal direction by choosing the best individual of the population as one of the parents and create variety by selecting other two random individuals as the rest of parents. The reason why there are two random parents rather than only one is that, given that one of the three parents is fixed, we are supposed to introduce more diverse edges to prevent the underlying homogeneity of the combination. The three-parent combination operator can make full use of high-quality edge sets obtained from distant searching regions during the search process. At each generation of the proposed CMA, the combination operator is applied to create a new offspring which is a candidate solution for population updating. The time complexity of the combination operator is O(n).

### 3.7. Mutation Operator

To prevent the population from premature and reinforce the global exploration capacity of the algorithm, a concise mutation operator is used when an offspring generated from the combination operator followed by local refinement is not diverse from existing solutions in the population. The mutation operator iteratively applies the unconstrained move operator (Section 3.4.1) for η times, where η obeys Lévy distribution [25] as the following Equation (Equation 15):(15)$$\eta =\frac{\lambda \Gamma (\lambda )\sin(\pi \lambda /2)}{\pi }\cdot \frac{1}{s^{1+\lambda }}\cdot n$$
where λ is the scaling factor, Γ(λ) is the standard gamma function and *s* is a random value taken within [0.2,0.4], applying the mutation operator every time. *n* is the number of vertices. Note that the move is chosen completely at random regardless of cost change or tabu status, and the procedure can be performed in O(n) time. The mutation operator can transition the current solution to a distant searching region, thus providing the population with a new member of the diversified feature.

### 3.8. Population Management

Diversity is a property of a group of individuals that indicates the degree these individuals are alike. An appropriate population update strategy is required to conserve population diversity during the search process, thereby preventing premature convergence and stagnation of the algorithm [26]. For each new solution *T* generated by the combination or mutation operator and returned by the tabu-based nearby exploration process, the proposed CMA adopts a concise management strategy to decide whether *T* updates the population under following rule. If *T* differs from any existing solution in the population, and is of better quality than the worst solution in the population, or the population has not been updated for 3 consecutive generations, then *T* replaces this worst member from the population; otherwise, it is discarded.

### 3.9. Computational Complexity of the Proposed Algorithm

The computational complexity of the proposed CMA mainly consists of two parts, including the initialization period and cycle period. In the initialization period, it takes $O(m^{2})$ for edge clustering and O(n) for population initialization, where where *m* is the number of edges and *n* is the number of edges, thus making the total complexity of initialization period $O(m^{2})$. In the cycle period, CMA executes three procedures: nearby exploration, combination and mutation. At each generation of CMA, the nearby exploration performs tabu search and local refinement in $O(n^{2}+n^{2})$, the combination operator generates an offspring solution in O(n) and the mutation operator generates a distinct solution in O(n). Thus, the total complexity of cycle period is $O(n^{2})$. To sum it up, the computational complexity of the proposed algorithm is $O(m^{2}+n^{2})$.

## 4. Computational Experiments

In this section, we perform a computational evaluation of the proposed memetic algorithm. We first describe the benchmark instances and the experimental settings. We then describe the edge clustering experiments and finally present the computational results obtained on the benchmark instances and compare them with the state-of-the-art algorithms.

### 4.1. Benchmark

In order to test the performance of the algorithm on the most challenging large-scale benchmark, the experiment adopts the benchmarks SS, RAND and SOAK with the largest number of vertices used in the literature. Each benchmark is described as follows:

The SS (Sundar et al. [15]) benchmark includes 18 complete graph instances with n= 25, 50, 100, 150, 200 and 250 (each corresponds to 3 instances). The linear integer costs are uniformly distributed at random within [1, 100], and the quadratic integer costs are randomly distributed in [1, 20].

The RAND (Lozano et al. [18]) benchmark includes 9 complete graph instances with n= 150, 200 and 250. For each *n*, there are 3 instances. The linear integer costs are uniformly distributed at random in [1, 100], and the quadratic integer costs between edges are randomly distributed in [1, 20].

The SOAK (Lozano et al. [18]) benchmark includes 9 complete graph instances with n= 150, 200 and 250. For each *n*, there are 3 instances. The vertices are distributed uniformly at random on a 500×500 grid. The linear costs are the integer Euclidean distance between any pair of vertices, and the quadratic costs are also integers uniformly distributed in [1, 20].

### 4.2. Experimental Settings

The proposed CMA is coded in C++ and compiled with a g++ compiler using option ‘-O3’. All experiments are performed on a computer with an AMD Ryzen 7 processor (2.9 GHz) and 16 GB RAM running under the Linux operating system.

As described in Section 3, the proposed CMA is controlled by several parameters (see Table 1). For different instances, the most suitable settings may be different; here, we give the settings tuned by experiments (Section 5.3) that can adapt to most instances as the default settings.

We fix $r=1.2,c_{\max}=4,\lambda =0.6$ and randomly take the value of $l_{e},I_{\mathrm{out}}$ and $p_{b}$ within the given intervals in every iteration during the search process. Cluster ratio *r* controls the size of a cluster relative to the number of its vertices; a larger *r* can provide more sufficient edges to build a tree while shrinking the population. Parameter $c_{\max}$ (max allowed iterations without improvement) is used to control the exploration strength: the larger it is, the more completely the nearby solution space is explored, along with more time cost. The exploration length $l_{e}$ decides whether the exploration can escape from local optima and to what extent it moves the current solution away from original position. The tabu tenures $I_{\mathrm{out}}$ are set to coordinate with $l_{e}$ to avoid detouring. Parameter $p_{b}$ determines the probability of selecting the best individual to provide edges for combination, a larger $p_{b}$ resulting in an offspring more similar to the best individual while losing diverse features from other parents. Parameter λ controls the mutation strength, a larger λ brings more diversity but may cause unnecessary time cost.

### 4.3. Edge Clustering Experiments

For the same cluster ratio *r*, the results obtained by running the clustering algorithm multiple times independently on the same instance are consistent; thus, we save the clustering results as known properties for each instance which is utilized at the solution initialization period in every run of CMA. Comparing different clustering results on all benchmarks, we found that the time cost and the number of clusters generated are almost the same among instances with the same |V|. Additionally, we also calculated the matching degree of the edge set of the best obtained solution on each instance with those in corresponding clusters, respectively, and recorded the highest of them as the hit rate. Then, we calculated the average hit rate for all instances with the same |V|. The above statistics are listed in Table 2, where columns 1–4 indicate the number of vertices |V|, the average time cost per run $t{\left(s\right)}_{\mathrm{avg}}$, the number of clusters generated $p_{\mathrm{avg}}$ and the average hit rate $r_{\mathrm{avg}}\left(\%\right)$. As shown in Table 2, we can finish clustering in considerable time and generate an adaptive quantity of clusters. Furthermore, we find that the cluster can match the real solution in a certain degree, which can provide the algorithm a promising initial solution in the population to start with. Moreover, we found that the hit rate decreases as the size of the instance increases, which may be owing to the increasing complexity and randomicity.

### 4.4. Experimental Results on Benchmark SS

Benchmark SS is used to evaluate three algorithms, including ABC [15], QMST-TS [8] and TPS [19]. Among them, we were given the code for TPS. Thus, for TPS, we list the results reported in [19], as well as the results obtained by running the TPS code on our computer under the same stopping condition, where each run stops if the best found solution cannot be further improved for 5 consecutive rounds, or up to 40 rounds have been applied. For ABC and QMST-TS, we quote the relevant results reported in the literature. For comparison, we execute CMA 20 times (like the comparing algorithm), with the stopping criterion set to 50 generations per execution. The above statistics are listed in Table 3. The first two columns of Table 3 are the number of vertices (|V|) and the index of each instance. The third column lists the four performance indicators: best objective value ($f_{\mathrm{best}}$), average objective value among 20 runs ($f_{\mathrm{avg}}$), standard deviations among 20 runs (std) and average execution time of each run t(s). Corresponding results of the comparison algorithm are listed from columns 4 to 8. The entries in bold indicate the best performance among all comparison algorithms, the symbol “*” indicates that the algorithm improves the current known optimal solution in the literature, and the entries with “-” indicate that the data do not exist.

It can be seen in Table 3 that the proposed CMA reports improved upper bounds and has obvious improvements in the average objective value and standard deviation on 12 instances with |V|>100, while matching the best performance on the remaining 6 small instances. Meanwhile, the proposed CMA is computationally more efficient than the reference algorithms, especially the state-of-the-art population-based algorithm ABC in the most challenging instances.

### 4.5. Experimental Results on Benchmark RAND and SOAK

The benchmarks RAND and SOAK are used to evaluate three algorithms, including ITS [3], HSII [18] and TPS [19]. These three algorithms execute 10 independent times on each instance within a time limit that varies according to problem size (400 s for n=150, 1200 s for n=200 and 2000 s for n=250). Like the experiments on the benchmark SS, we list the results reported in [19], as well as the results obtained by running the TPS code on our computer under the same stopping condition. For ITS and HSII, we quote the relevant results reported in the literature. For a fair comparison, we also perform the CMA 10 times on each case, with the same time limit as used in the above methods. The statistics are listed in Table 4 and Table 5, which are presented in the same paradigm as in Table 3 (Section 4.4).

It can be seen in Table 4 and Table 5 that the proposed CMA reports improved upper bounds on 13 out of 18 instances and matches the best-known results on 3 SOAK instances, while only being outperformed by TPS on 2 RAND instances, where TPS is further retuned. In terms of average objective value and standard deviation, CMA exhibits a better performance compared with the three reference algorithms on all 18 challenging instances. This demonstrates the satisfactory competence of CMA in terms of both solution quality and searching stability compared with the state-of-the-art QMSTP methods.

## 5. Analyses and Discussions

In this section, we conduct further analysis to investigate the effectiveness of the edge clustering mechanism (Section 3.3) and the three-parent combination operator (Section 3.6). Then, we provide a sensitivity analysis for the optimal parameter (Section 4.2) selection of CMA. Finally, we give the discussions of this work.

### 5.1. Impact of Edge Clustering on Convergence Speed

In order to illustrate the impact of edge clustering on convergence speed, we compare standard CMA with its weakened version (denoted as CMA*), where the edge clustering mechanism guided initialization procedure is disabled and replaced by the conventional initialization procedure. Specifically, CMA* generates each initial solution from the whole edge set *E* of graph G=(V,E), instead of a corresponding cluster *C* generated by edge clustering. Starting from an empty solution with only root vertex, CMA* iteratively selects a legal edge *e* from *E* to construct an initial tree *T*, where edge *e* meets the requirement described in Section 3.5.2. CMA* generates the same amount of initial solutions and maintains the same subsequent searching procedures as those in CMA for fairness of comparison. Furthermore, we also conduct the state-of-the-art algorithm TPS [19] to reveal the competitiveness of CMA*, which stands for the basic framework of our memetic algorithm.

We chose four challenging instances with n=150 (RAND150-1, SOAK150-1) and n=250 (RAND250-1, SOAK250-1) to evaluate the convergence speed of the mentioned algorithm. In this experiment, we run CMA, CMA* and TPS 10 times on each instance, respectively. The execution time for each run is set to 300 s for instances with n=150 and 1000 s for instances with n=250. We record the obtained best cost every 5 s for instances with n=150 and every 10 s for instances with n=250 during the searching process. The CMA and CMA* follow the settings in Section 4.2, and the TPS follows the settings in the literature. Figure 5 shows the running profiles of CMA, CMA* and TPS on the chosen instances, where the X-axis represents the running time and the Y-axis shows the average cost over 10 runs.

It can be seen from Figure 5 that there is a clear performance deterioration, especially in the early searching stage, without the guidance of the edge clustering mechanism in the population initialization, indicating that edges can be grouped with short “distance” (total cost) in the same cluster, which makes the initial solutions closer to high-quality search regions and more likely to be complemented by each other. Indeed, with the conventional initialization alone, CMA* is already a competent algorithm when compared with the reference TSP algorithm, where CMA* can continue to explore the solution space without an early come stagnation, showing a favorable search behavior of memetic framework. When considering the extra time consumption brought by the edge clustering mechanism, it is also a considerable choice to run CMA without clustering.

### 5.2. Impact of Three-Parent Combination Operator

To investigate the effectiveness of the three-parent combination operator, which is a key component of the proposed CMA, we compared two versions of CMA with and without the combination operator on 18 challenging instances from the benchmarks RAND and SOAK. We performed the two versions of CMA on each instance 10 times with the corresponding execution times of 300 s for n=150, 600 s for n=200 and 1000 s for n=250 per run. The other searching component and parameter are the defaults, as described in Section 4.2. The results are shown in Figure 6, where the Y-axis represents the %gap between the optimal result (see Table 4 and Table 5) and the average result obtained over 10 runs on each instance. The comparison shows that CMA with the combination operator obtains an average advantage over CMA without the operator on 18 out of 18 challenging instances, which constitutes a strong motivation for the use of the three-parent combination operator.

### 5.3. Parameter Sensitivity Analysis

We analyze the sensitivity of the four CMA parameters: max allowed iterations without improvement $c_{\max}$, the exploration length $l_{e}$, probability of parent selecting $p_{b}$ and mutation parameter λ. To evaluate the sensitivity of each parameter, we test the varying considered values for each parameter, while fixing the other parameters to their default value (see Table 2). For this analysis, we use the same selection of 18 instances as Section 5.2 and perform 10 runs for each considered value. The execution cutoff time per run is the same as specified in Section 5.2. Figure 7 shows the box plots of the experimental results, where the X-axis indicates the parameter values and the Y-axis shows the distribution and range of the normalized average results. To determine whether there exists a statistically significant difference in solution samples for different values of a given parameter, we employ the Friedman rank sum test. The corresponding *p*-values are also given in Figure 7. The Friedman test reveals a statistically significant difference in performance for $c_{\max}$ (*p*-value = 0.0114), $l_{e}$ (*p*-value = $9.0962\times 10^{-4}$), $p_{b}$ (*p*-value = $1.5690\times 10^{-4}$) and λ (*p*-value = 0.0064). From the plots in Figure 7, we determine the best-performing parameter settings as the default settings, which are described in Section 4.2.

### 5.4. Discussions

As mentioned in Section 1, metaheuristic algorithms for QMSTP mainly consist of two categories, including population-based algorithms (e.g., artificial bee colony algorithm [15] and hybrid genetic algorithm [3]) and local search algorithms (e.g., iterative tabu search [18] and local search [16]). As described above, CMA incorporates the merits of both the population-based exploration and the local-based intensification, which contributes to its competitiveness with the state-of-the-art methods in the literature. Precisely, the competence of the proposed CMA is attributed to the integration of three distinguishing elements. First, CMA applies a clustering mechanism to the edges of the graph, where edges can be grouped with short “distance” (total cost) in the same cluster. This mechanism can effectively guide the population initialization. As illustrated in Section 5.1, the applied edge clustering mechanism effectively enhances the convergence speed at an early stage of the proposed CMA. Second, CMA adopts a tabu-based nearby exploration phase to explore the restricted surrounding areas to locate other local optima. It combines an iterative tabu search with an efficient local refinement procedure and a revisiting mechanism to guarantee the search intensification around chosen high-quality local optima. Last but not least, CMA employs a new combination operator and a new mutation operator. By inheriting valuable edges, the combination operator can generate promising and diversified offspring solutions that are used as starting points for local refinement. As uncovered in Section 5.2, the dedicated combination operator significantly strengthens the search capability of the proposed CMA. The mutation operator, which is designed based on Lévy distribution, can explore unvisited search regions, and it can prevent the population from premature. Theses problem-specific components are combined into a whole, where all parts coordinate and promote each other to find the optima.

In addition, as demonstrated by parameter sensitivity analysis in Section 5.3, the performance of CMA is primarily controlled by parameters $c_{\max}$, $l_{e}$, which determine the intensity of surrounding exploitation, and parameters $p_{b}$, λ, which ensure the diversity of new starting points, bringing about a trade-off between global and local search.

## 6. Conclusions

The quadratic minimum spanning tree problem is a fundamental problem in network topology design. To tackle this problem, we present for the first time a clustering-enhanced memetic algorithm (CMA) that combines an edge clustering mechanism to guide the population initialization, a tabu-based nearby exploration phase to search nearby local optima in restricted area, a three-parent combination operator to generate promising offspring solutions and a mutation operator based on Lévy distribution to prevent the population from premature. Computational experiments on 3 sets of 36 benchmark instances validate the competitiveness of the proposed CMA compared with the state-of-art approaches in terms of the solution quality, convergence speed and robustness. In particular, CMA reports improved upper bounds for the 25 most challenging benchmark instances with unproven optimum, while matching the best-known results for all but 2 remaining instances. Additional experiments verify the positive contribution of edge clustering mechanism and combination operator based on edge set disruption-recombination to the performance of CMA. Furthermore, we investigate the sensitivity and tune the best selection of major parameters.

### 6.1. Limitations

Although the proposed algorithm has good performance, there are still some limitations. First, the edge clustering mechanism should be further integrated with the search process to strengthen the utilization of clustering results. Second, the population management strategy is overly simplistic. Other population management strategies presented in the memetic algorithm literature could be considered. Finally, the exploration operators used in this work are all based on a 1-exchange move operator. It might be useful to consider an *N*-exchange move operator and corresponding evaluation techniques which can replace *N* edges in the meantime, while keeping the remaining parts of the tree unchanged.

### 6.2. Future Research Perspectives

First, it would be worthy to consider adaptations of CMA to other problems related to QMSTP, e.g., the adjacent only quadratic minimum spanning tree problem [4,27], the rough–fuzzy quadratic minimum spanning tree problem [5] and uncertain quadratic minimum spanning tree problem [28]. Second, as feasible and infeasible searches have shown strong performances [29,30] in combinatorial optimization problems, it would be beneficial to consider feasible and infeasible search mechanisms for an accelerated exploration of the search space with CMA. Finally, in recent years, there have been many efforts to improve optimization approaches using machine learning techniques [31,32]. This work enriches the study of the use of agglomerative clustering mechanisms to solve optimization problems more efficiently.

## Figures and Tables

**Figure 1 entropy-25-00087-f001:**
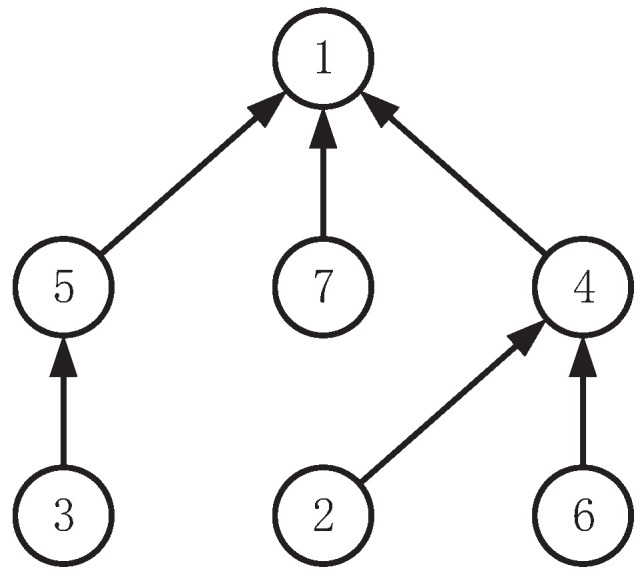
Tree and its representation.

**Figure 2 entropy-25-00087-f002:**
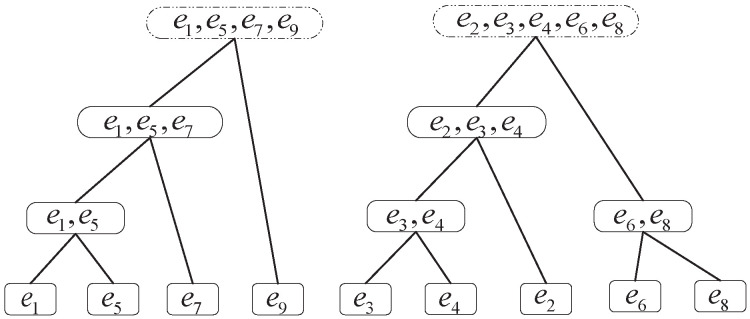
Agglomerative clustering for an example with 9 edges.

**Figure 3 entropy-25-00087-f003:**
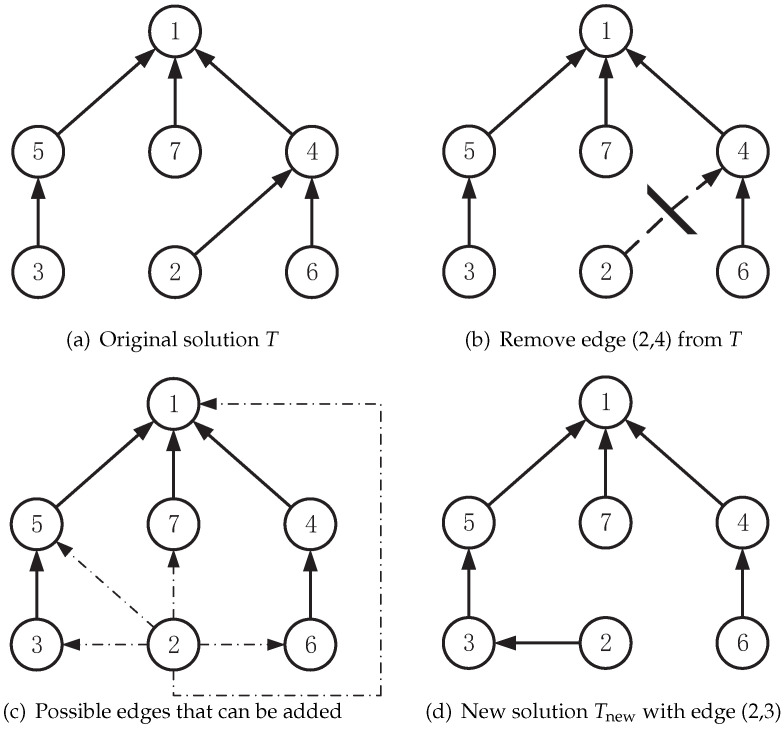
Process of the move operator.

**Figure 4 entropy-25-00087-f004:**
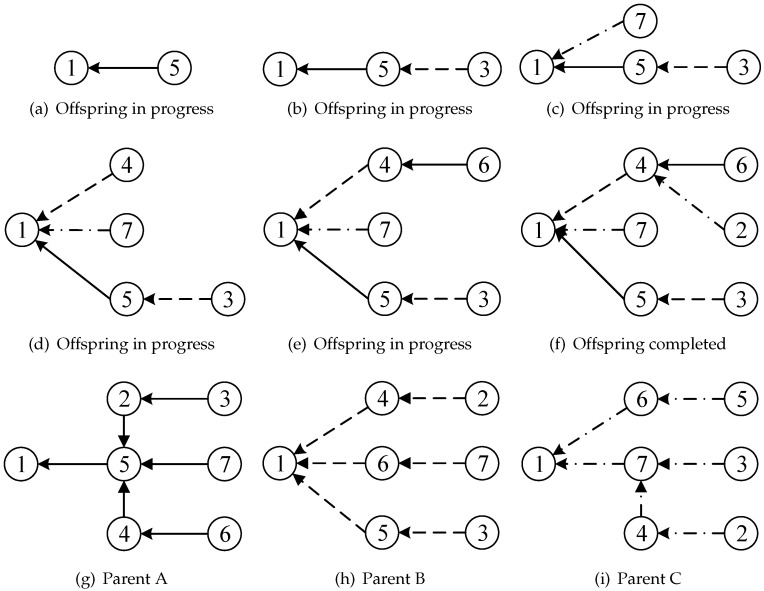
Offspring solution generated by three-parent combination operator. (**a**–**f**) are the process of offspring generation; (**g**–**i**) are the parents used by the generation above.

**Figure 5 entropy-25-00087-f005:**
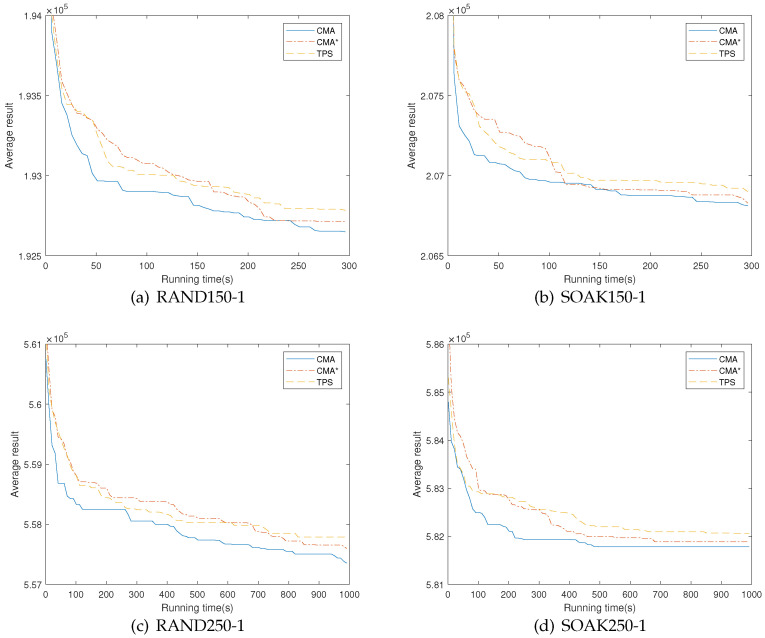
Running profiles of standard CMA and CMA* (a version of CMA without edge clustering) for solving RAND150-1, SOAK150-1, RAND250-1 and SOAK250-1.

**Figure 6 entropy-25-00087-f006:**
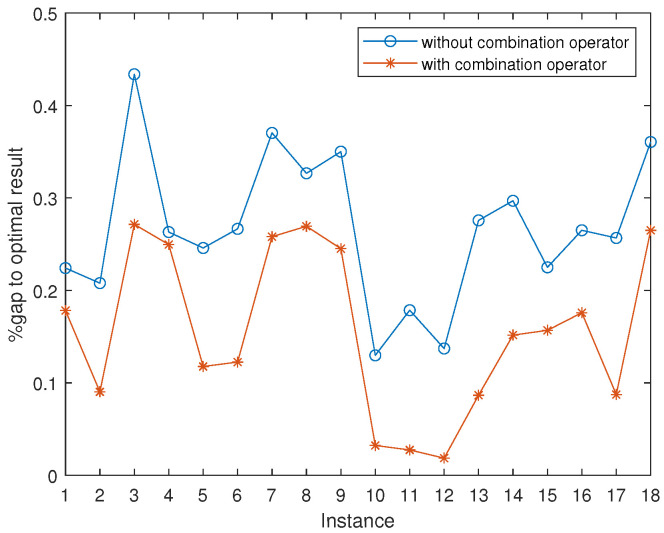
Comparisons of CMA with its variant that excludes the three-parent combination operator.

**Figure 7 entropy-25-00087-f007:**
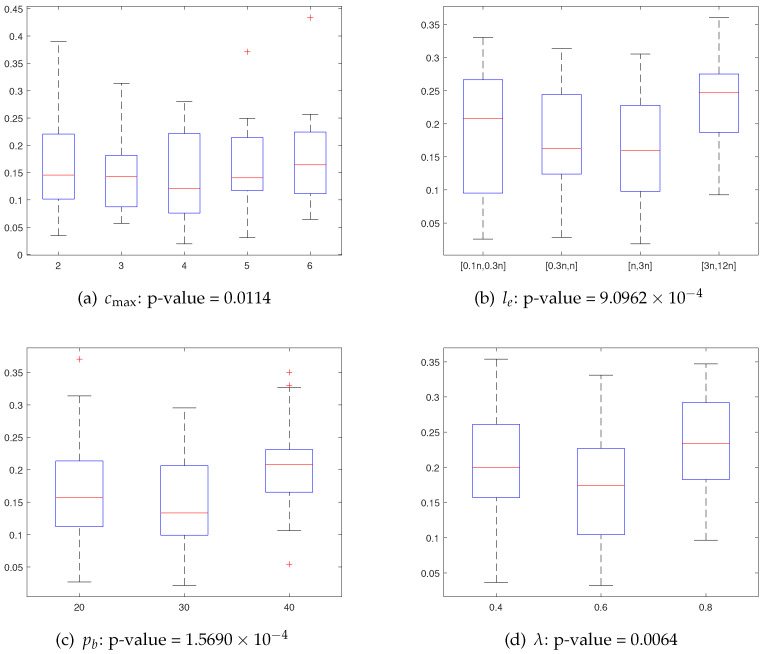
Boxplots of the normalized average objective values for each considered value of a given parameter.

**Table 1 entropy-25-00087-t001:** Setting of parameters.

Parameter	Section	Description	Default Value
*r*	Section 3.3.1	cluster ratio	1.2
$c_{\max}$	Section 3.5	max allowed iterations without improvement	4
$l_{e}$	Section 3.5.1	exploration length	[n,3n]
$I_{\mathrm{out}}$	Section 3.5.2	tabu tenures	[n,3n]
$p_{b}$	Section 3.6	probability of parent selecting	[20,40]
λ	Section 3.7	mutation parameter	0.6

**Table 2 entropy-25-00087-t002:** Results of clustering.

|V|	$t{\left(s\right)}_{\mathrm{avg}}$	$p_{\mathrm{avg}}$	$r_{\mathrm{avg}}\;\left(\%\right)$
25	0.002	7	63.4
50	0.03	15	57.7
100	0.9	29	52.7
150	7	46	48.2
200	30	61	44.8
250	98	66	40.5

**Table 3 entropy-25-00087-t003:** Experimental results of SS instances.

|V|	Index	Results	ABC [15]	QMTS-TS [8]	TPS [19]	TPS *	CMA
25	1	$f_{\mathrm{best}}$	**5085**	**5085**	**5085**	**5085**	**5085**
		$f_{\mathrm{avg}}$	5085.85	-	-	**5085**	**5085**
		std	3.71	-	-	0	0
		t(s)	0.91	0.33	0.02	0.02	0.04
	2	$f_{\mathrm{best}}$	**5081**	**5081**	**5081**	**5081**	**5081**
		$f_{\mathrm{avg}}$	5101.2	-	-	**5081**	**5081**
		std	6.64	-	-	0	0
		t(s)	1.02	0.33	0.03	0.04	0.04
	3	$f_{\mathrm{best}}$	**4962**	**4962**	**4962**	**4962**	**4962**
		$f_{\mathrm{avg}}$	**4962**	-	-	**4962**	**4962**
		std	0	-	-	0	0
		t(s)	1.05	0.35	0.02	0.02	0.04
50	1	$f_{\mathrm{best}}$	**21,126**	**21,126**	**21,126**	**21,126**	**21,126**
		$f_{\mathrm{avg}}$	21,157.25	-	-	**21,126**	**21,126**
		std	34.4	-	-	0	0
		t(s)	8.68	2.52	0.28	0.25	0.3
	2	$f_{\mathrm{best}}$	21,123	**21,106**	**21,106**	**21,106**	**21,106**
		$f_{\mathrm{avg}}$	21,179.85	-	-	**21,106**	**21,106**
		std	32.47	-	-	0	0
		t(s)	8.84	2.52	0.28	0.41	0.3
	3	$f_{\mathrm{best}}$	21,059	**21,059**	**21,059**	**21,059**	**21,059**
		$f_{\mathrm{avg}}$	21,091.95	-	-	**21,059**	**21,059**
		std	59.67	-	-	0	0
		t(s)	9.51	2.53	0.21	0.23	0.3
100	1	$f_{\mathrm{best}}$	89,098	88,871	88,745	88,944	**88,701**
		$f_{\mathrm{avg}}$	89,404.6	-	-	89,086.56	**88,760.89**
		std	167.87	-	-	41.65	20.8
		t(s)	116.66	48.29	4.06	8.65	6.55
	2	$f_{\mathrm{best}}$	89,202	89,049	88,911	88,894	**88,843**
		$f_{\mathrm{avg}}$	89,520.45	-	-	88,991.78	**88,866.11**
		std	190.05	-	-	63.62	26.58
		t(s)	115.95	47.88	3.82	7.42	6.72
	3	$f_{\mathrm{best}}$	89,007	88,720	88,659	88,658	**88,627**
		$f_{\mathrm{avg}}$	89,242.6	-	-	88,872.89	**88,735.33**
		std	138.59	-	-	196.38	73.52
		t(s)	98.88	48.06	4.38	8.78	6.14
150	1	$f_{\mathrm{best}}$	205,619	205,615	204,995	205,083	**204,937 ***
		$f_{\mathrm{avg}}$	206,404.3	-	-	205,426.11	**205,168**
		std	405.97	-	-	197.99	90.87
		t(s)	444.87	146.44	25.53	40.77	34.79
	2	$f_{\mathrm{best}}$	205,874	205,509	205,219	205,185	**205,034 ***
		$f_{\mathrm{avg}}$	206,300.55	-	-	205,404.22	**205,183.56**
		std	243.36	-	-	142.27	55.62
		t(s)	374.33	146.15	24.69	38.11	35.86
	3	$f_{\mathrm{best}}$	205,634	205,094	205,076	205,055	**205,028 ***
		$f_{\mathrm{avg}}$	206,160.1	-	-	205,406.67	**205,196.78**
		std	316.07	-	-	265.88	125.89
		t(s)	432.93	146.43	29.84	31.35	32.77
200	1	$f_{\mathrm{best}}$	371,797	371,492	370,873	370,818	**370,715 ***
		$f_{\mathrm{avg}}$	372,527.6	-	-	371,215.22	**371,008.33**
		std	381.44	-	-	312.33	183.43
		t(s)	1141.42	316.02	75.98	69.28	57.89
	2	$f_{\mathrm{best}}$	371,864	371,698	370,853	370,825	**370,824 ***
		$f_{\mathrm{avg}}$	372,306.6	-	-	371,472.11	**371,170.33**
		std	311.74	-	-	395.34	194.92
		t(s)	1155.6	316.61	69.32	59.83	55.44
	3	$f_{\mathrm{best}}$	372,156	371,584	370,954	370,943	**370,901 ***
		$f_{\mathrm{avg}}$	372,842.9	-	-	371,206	**371,085.67**
		std	735.21	-	-	191.9	146.81
		t(s)	1276.71	316.22	64.14	64.26	56.21
250	1	$f_{\mathrm{best}}$	587,924	586,861	586,265	586,196	**586,171 ***
		$f_{\mathrm{avg}}$	588,785.1	-	-	586,809	**586,718.67**
		std	578.65	-	-	489.5	315.59
		t(s)	2563.41	478.62	145.83	151.16	124.8
	2	$f_{\mathrm{best}}$	588,068	587,607	586,778	586,757	**586,514 ***
		$f_{\mathrm{avg}}$	588,731.45	-	-	586,907.44	**586,782.89**
		std	368.08	-	-	272.22	181.11
		t(s)	2840.91	479.64	113.17	140.53	126.7
	3	$f_{\mathrm{best}}$	587,883	587,281	585,851	585,851	**585,783 ***
		$f_{\mathrm{avg}}$	588,534.95	-	-	586,683.89	**586,415.44**
		std	463.2	-	-	550.31	329.49
		t(s)	2328.29	480.06	135.5	157.12	118.9

**Table 4 entropy-25-00087-t004:** Experimental results of RAND instances.

|V|	Index	Results	ITS [3]	HSII [18]	TPS [19]	TPS *	CMA
150	1	$f_{\mathrm{best}}$	192,946	192,606	192,369	192,427	**192,329 ***
		$f_{\mathrm{avg}}$	193,244.5	192,910.1	-	192,688.8	**192,576.3**
		std	-	-	-	157.08	126.07
		t(s)	400	400	400	400	400
	2	$f_{\mathrm{best}}$	193,034	192,607	192,579	192,558	**192,460 ***
		$f_{\mathrm{avg}}$	193,369.9	192,922.8	-	192,702.8	**192,628.1**
		std	-	-	-	135.65	129.37
		t(s)	400	400	400	400	400
	3	$f_{\mathrm{best}}$	192,965	192,577	192,046	192,269	**192,008 ***
		$f_{\mathrm{avg}}$	193,303.1	192,792.6	-	192,657.8	**192,490.1**
		std	-	-	-	193.34	163.36
		t(s)	400	400	400	400	400
200	1	$f_{\mathrm{best}}$	351,216	350,517	350,321	350,394	**350,297 ***
		$f_{\mathrm{avg}}$	351,787.2	351,023.6	-	350,878.8	**350,724**
		std	-	-	-	201.72	158.01
		t(s)	1200	1200	1200	1200	1200
	2	$f_{\mathrm{best}}$	351,312	350,389	**350,231**	350,576	350,446
		$f_{\mathrm{avg}}$	351,823.7	350,902.4	-	350,918.4	**350,750.6**
		std	-	-	-	236.06	163.72
		t(s)	1200	1200	1200	1200	1200
	3	$f_{\mathrm{best}}$	351,466	351,057	350,601	350,677	**350,544 ***
		$f_{\mathrm{avg}}$	351,940.8	351,285.4	-	350,929.2	**350,809.6**
		std	-	-	-	243.37	182.45
		t(s)	1200	1200	1200	1200	1200
250	1	$f_{\mathrm{best}}$	558,451	556,929	556,596	556,897	**556,588 ***
		$f_{\mathrm{avg}}$	559,235.5	557,434.6	-	557,689.4	**557,363.2**
		std	-	-	-	352.75	326.27
		t(s)	2000	2000	2000	2000	2000
	2	$f_{\mathrm{best}}$	558,820	557,474	556,604	556,823	**556,598 ***
		$f_{\mathrm{avg}}$	559,478.2	557,850.1	-	557,571.8	**557,356.6**
		std	-	-	-	389	315.1
		t(s)	2000	2000	2000	2000	2000
	3	$f_{\mathrm{best}}$	559,304	556,813	**556,378**	557,014	556,610
		$f_{\mathrm{avg}}$	559,489.8	557,463.4	-	557,456.3	**557,204.5**
		std	-	-	-	339.73	328.36
		t(s)	2000	2000	2000	2000	2000

**Table 5 entropy-25-00087-t005:** Experimental results of SOAK instances.

|V|	Index	Results	ITS [3]	HSII [18]	TPS [19]	TPS *	CMA
150	1	$f_{\mathrm{best}}$	206,721	206,925	**206,721**	**206,721**	**206,721**
		$f_{\mathrm{avg}}$	207,004.9	207,089.4	-	206,850	**206,795.1**
		std	**-**	**-**	-	123.08	98.09
		t(s)	400	400	400	400	400
	2	$f_{\mathrm{best}}$	206,761	207,102	**206,761**	**206,761**	**206,761**
		$f_{\mathrm{avg}}$	207,153.6	207,280.5	-	206,889.9	**206,853.9**
		std	**-**	**-**	-	105.8	83.43
		t(s)	400	400	400	400	400
	3	$f_{\mathrm{best}}$	206,802	206,781	**206,759**	**206,759**	**206,759**
		$f_{\mathrm{avg}}$	206,959.6	206,954.2	-	206,847.5	**206,827.1**
		std	**-**	**-**	-	87.49	59.3
		t(s)	400	400	400	400	400
200	1	$f_{\mathrm{best}}$	370,137	370,265	369,851	369,851	**369,840 ***
		$f_{\mathrm{avg}}$	370,533.3	370,530.5	-	370,111.8	**370,032.1**
		std	**-**	**-**	-	175.59	138.84
		t(s)	1200	1200	1200	1200	1200
	2	$f_{\mathrm{best}}$	370,028	369,982	369,803	369,835	**369,754 ***
		$f_{\mathrm{avg}}$	370,351	370,183.6	-	370,125.6	**370,058.2**
		std	**-**	**-**	-	201.3	143.47
		t(s)	1200	1200	1200	1200	1200
	3	$f_{\mathrm{best}}$	370,046	370,045	369,794	369,809	**369,705 ***
		$f_{\mathrm{avg}}$	370,390.7	370,345.8	-	370,080	**369,918.3**
		std	**-**	**-**	-	189.65	147.91
		t(s)	1200	1200	1200	1200	1200
250	1	$f_{\mathrm{best}}$	582,282	581,819	581,671	581,543	**581,439 ***
		$f_{\mathrm{avg}}$	583,069.4	582,283.7	-	581,933.2	**581,777.2**
		std	**-**	**-**	-	236.16	165.84
		t(s)	2000	2000	2000	2000	2000
	2	$f_{\mathrm{best}}$	582,145	581,691	581,492	581,463	**581,364 ***
		$f_{\mathrm{avg}}$	582,872.9	582,013.3	-	581,886.6	**581,752.1**
		std	**-**	**-**	-	270.95	204.35
		t(s)	2000	2000	2000	2000	2000
	3	$f_{\mathrm{best}}$	582,708	581,854	581,573	581,449	**581,207 ***
		$f_{\mathrm{avg}}$	583,525.8	582,590.8	-	581,952.3	**581,837.4**
		std	**-**	**-**	-	309.75	248.84
		t(s)	2000	2000	2000	2000	2000

## Data Availability

All the benchmark instances can be provided on request to the author (zsfshufan@163.com).

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
