# Peer review of "A Clustering-Enhanced Memetic Algorithm for the Quadratic Minimum Spanning Tree Problem"

_entropy, 2022, doi:10.3390/e25010087_

Round 1

Reviewer 1 Report

The authors have first presented a clustering enhanced memetic algorithm for the quadratic minimum spanning tree problem. The manuscript has made some contributions. However, the entire structure of the manuscript needs to be further improved. Several major and minor amendments are required as follows:

1.      The introduction needs to be further strengthened. In addition, the author's contributions can be clearly listed in the form of highlights.

2.      In algorithm comparison, how does this manuscript ensure the fairness of the algorithm? The fitness calculation is called many times in the local thinning process. Is there such an operation in other comparison algorithms?

3.      Please add the time complexity analysis of the method in the corresponding position.

4.      CMA * needs to be further clarified in the manuscript.

5.      What are your criteria for evaluating the advantages of an algorithm? It is suggested to use some mathematical statistics methods to judge, not just the average and best value.

6.      The ABC algorithm in the manuscript is not used in SOAK and other examples. I don't know if it was lost or for some other reason. In addition, some mainstream methods in recent years need to be added for comparison.

7.      Please further improve the content of the optimization convergence diagram. Compared with optimization methods, a large number of rich convergence evolution graphs will appear to give readers an intuitive view of the progressiveness of your algorithm.

8.      The manuscript needs further polishing.

Reviewer 2 Report

There are many obscure notations which make the paper difficult to read.

The followings are my comments:

There is no clear explanation of $V$, $E$, $c_e$ and $q_{ef}$ in

Section 2.  Since the problem description is unclear, it is difficult

to understand the entire paper.

You need a little more explanation on Eq. 5.

What is $w_e$ ? Is there some relation between $w_e$ and $c_e$ in Eq. 1 ?

What do you mean by $D_{min},a,b \leftarrow min D_{ij}$ in Algorithm 2 ?

The explanation on Figure 2 (from line 149 to 151) is unclear.

Could you explain this example in more detail ?

In Figure 4, it is unclear how to get from (a) to (f).

I need more explanation on Figure 4.

Typos:

line 90 "assume all, and all.": What do you mean by this sentence ?

line 125: "agglomeritive" -> "agglomerative" ?

line 175: "Eq. ??"

line 178: "the the" -> "the"

Page 8, before Eq. 13: What is $m$ ?

Page 8, Eq. 14: $q_{gj}$ -> $q_{gf}$ ?

Algorithm 5, lines 7 to 9: Identical to lines 11 to 13.

line 286: $c_{max} = 5$ and $\lambda = 6$ are different from that in Table 1.

line 308: "2]" -> "2"

line 355: "annlysis" -> "analysis"

Reviewer 3 Report

The presented paper does not clearly declare the novelty. There are many experiments and tables, however, it lacks the proper discussion. There are no conclusion section pointing to the limitations, as well as future research perspectives. Instead, just general (and also well known) sentences are stated. 

Looking to the references, there are many outdated references. 

The paper lacks the complex evaluation comparing existing solutions to declare the performance and novelty. 

Based on the consideration of the paper quality, unfortunately, I cannot recommend the paper for the publication. 

Round 2

Reviewer 1 Report

The author's response has met most of my questions. Compared with the original, the quality of the revised version has been further improved. I think we can accept it in this journal.

Reviewer 2 Report

Typos:

line 192: "{e_3}" -> "{e_4}"

line 192 "{e_1,e_5} and {e_7}": Duplicate

Reviewer 3 Report

The authors have applied the comments and recommendations. 
